# Higher-order interactions shape collective dynamics differently in hypergraphs and simplicial complexes

Yuanzhao Zhang ●[1,5] ✉, Maxime Lucas ●[2,3,5] ✉ & Federico Battiston ●[4] ✉

Higher-order networks have emerged as a powerful framework to model complex systems and their collective behavior. Going beyond pairwise interactions, they encode structured relations among arbitrary numbers of units through representations such as simplicial complexes and hypergraphs. So far, the choice between simplicial complexes and hypergraphs has often been motivated by technical convenience. Here, using synchronization as an example, we demonstrate that the effects of higher-order interactions are highly representation-dependent. In particular, higher-order interactions typically enhance synchronization in hypergraphs but have the opposite effect in simplicial complexes. We provide theoretical insight by linking the synchronizability of different hypergraph structures to (generalized) degree heterogeneity and cross-order degree correlation, which in turn influence a wide range of dynamical processes from contagion to diffusion. Our findings reveal the hidden impact of higher-order representations on collective dynamics, highlighting the importance of choosing appropriate representations when studying systems with nonpairwise interactions.

For the past three decades, networks have been successfully used to model complex systems with many interacting units. In their traditional form, networks only encode pairwise interactions[1,2]. Growing evidence, however, suggests that a node may often experience the influence of multiple other nodes in a nonlinear fashion and that such higher-order interactions cannot be decomposed into pairwise ones[3–7]. Examples can be found in a wide variety of domains including human dynamics[8], collaborations[9], ecological systems[10], and the brain[11,12]. Higher-order interactions not only impact the structure of these systems[13–21], they also often reshape their collective dynamics[22–27]. Indeed, they have been shown to induce novel collective phenomena, such as explosive transitions[28], in a variety of dynamical processes including diffusion[29,30], consensus[31,32], spreading[33–35], and evolution[36].

Despite many recent theoretical advances[37–42], little attention has so far been given to how higher-order interactions are best represented[43]. There are two mathematical frameworks that are most commonly used to model systems with higher-order interactions: hypergraphs[44] and simplicial complexes[45]. In most cases, the two representations have been used interchangeably and the choice for one or the other often appears to be motivated by technical convenience. For example, topological data analysis[46] and Hodge decomposition[47] require simplicial complexes. Here, we ask: Are there hidden consequences of choosing one higher-order representation over the other that could significantly impact the collective dynamics?

Answering this question is important given that, currently, reliable real-world hypergraph data are still scarce (with most existing ones concentrated in social systems), especially for complex dynamical systems such as the brain. For these systems, in order to study the effect of higher-order interactions, we have to start from data on pairwise networks and infer the potential nonpairwise connections[48]. A popular practice is to assume homophily between pairwise and

[1]Santa Fe Institute, Santa Fe, NM, USA. [2]ISI Foundation, Torino, Italy. [3]CENTAI Institute, Torino, Italy. [4]Department of Network and Data Science, Central European University, Vienna, Austria. [5]These authors contributed equally: Yuanzhao Zhang, Maxime Lucas. ✉e-mail: yzhang@santafe.edu; ml.maximelucas@gmail.com; battistonf@ceu.edu

nonpairwise interactions (e.g., by attaching three-body interactions to closed triangles in the network), effectively choosing simplicial complex as the higher-order representation. However, if different ways of adding hyperedges can fundamentally change the collective dynamics, then conclusions drawn from investigating a single higher-order representation could be misleading.

To explore this issue, we focus on synchronization—a paradigmatic process for the emergence of order in populations of interacting entities. It underlies the function of many natural and man-made systems[49,50], from circadian clocks[51] and vascular networks[52] to the brain[53]. Nonpairwise interactions arise naturally in synchronization from the phase reduction of coupled oscillator populations[54–58]. A key question regarding higher-order interactions in this context is: When do they promote synchronization? Recently, hyperedge-enhanced synchronization has been observed for a wide range of node dynamics[39,41,59–61]. It is thus tempting to conjecture that nonpairwise interactions synchronize oscillators more efficiently than pairwise ones. This seems physically plausible given that higher-order interactions enable more nodes to exchange information simultaneously, thus allowing more efficient communication and ultimately leading to enhanced synchronization performance.

In this article, we show that whether higher-order interactions promote or impede synchronization is highly representation-dependent. In particular, through a rich-get-richer effect, higher-order interactions consistently destabilize synchronization in simplicial complexes. On the other hand, higher-order interactions tend to stabilize synchronization in a broad class of hypergraphs, including random hypergraphs and semi-structured hypergraphs constructed from synthetic networks as well as brain connectome data. Offering a theoretical underpinning for the representation-dependent synchronization performance, we link the opposite trends to the different higher-order degree heterogeneities under the two representations. Furthermore, we investigate the impact of cross-order degree correlations for different families of hypergraphs. Since degree heterogeneity and degree correlation play a key role not only in synchronization but also in other dynamical processes such as diffusion and contagion, the effect of higher-order representations discovered here is likely to be crucial in complex systems beyond coupled oscillators.

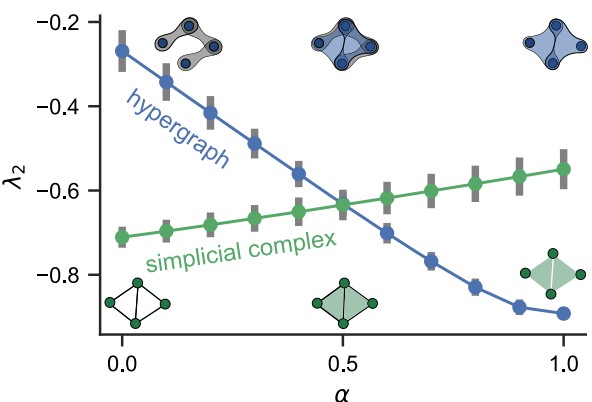

**Fig. 1 | Synchronization is enhanced by higher-order interactions in random hypergraphs but is impeded in simplicial complexes.** The maximum transverse Lyapunov exponent $\lambda_2$ is plotted against $\alpha$ for random hypergraphs (blue) and simplicial complexes (green). As $\alpha$ is increased, the coupling goes from first-order-only ($\alpha = 0$) to second-order-only ($\alpha = 1$). Each point represents the average over 50 independent hypergraphs or simplicial complexes with $n = 100$ nodes. The error bars represent standard deviations. We set $p = p_\triangle = 0.1$ for random hypergraphs and $p = 0.5$ for simplicial complexes.

## Results

### Higher-order interactions hinder synchronization in simplicial complexes but facilitate it in random hypergraphs

To isolate the effect of higher-order interactions from node dynamics, we consider a simple system consisting of $n$ identical phase oscillators[39], whose states $\boldsymbol{\theta} = (\theta_1, \cdots, \theta_n)$ evolve according to

$$\dot{\theta}_i = \omega + \frac{\gamma_1}{\langle k^{(1)} \rangle} \sum_{j=1}^{n} A_{ij} \sin(\theta_j - \theta_i) + \frac{\gamma_2}{\langle k^{(2)} \rangle} \sum_{j,k=1}^{n} \frac{1}{2} B_{ijk} \frac{1}{2} \sin(\theta_j + \theta_k - 2\theta_i). \quad (1)$$

Equation (1) is a natural generalization of the Kuramoto model[62] that includes interactions up to order two (i.e., three-body interactions). The oscillators have natural frequency $\omega$ and the coupling strengths at each order are $\gamma_1$ and $\gamma_2$, respectively. The adjacency tensors determine which oscillators interact: $A_{ij} = 1$ if nodes $i$ and $j$ have a first-order interaction, and zero otherwise. Similarly, $B_{ijk} = 1$ if and only if nodes $i$, $j$ and $k$ have a second-order interaction. All interactions are assumed to be unweighted and undirected. The (generalized) degrees are given by $k_i^{(1)} = \sum_{j=1}^{n} A_{ij}$ and $k_i^{(2)} = \frac{1}{2} \sum_{j,k}^{n} B_{ijk}$, respectively. Here, we normalize $B_{ijk}$ by a factor of two to avoid double counting the same 2-simplex.

Following refs. [59,63], we set

$$\gamma_1 = 1 - \alpha, \quad \gamma_2 = \alpha, \quad \alpha \in [0,1]. \quad (2)$$

The parameter $\alpha$ controls the relative strength of the first- and second-order interactions, from all first-order ($\alpha = 0$) to all second-order ($\alpha = 1$), allowing us to keep the total coupling budget constant and compare the effects of pairwise and non-pairwise interactions fairly. In addition, we normalize each coupling strength by the average degree of the corresponding order, $\langle k^{(\ell)} \rangle$.

Finally, we normalize the second-order coupling function by an additional factor of two so that each interaction contributes to the dynamics with an equal weight regardless of the number of oscillators involved. Note that another interaction term of the form $\sin(2\theta_j - \theta_k - \theta_i)$ appears naturally in other formulations obtained from phase reduction[54,55,57]. This type of term was shown to be dynamically equivalent to that in Eq. (1) when considering the linearized dynamics around full synchrony[39]. Indeed, they yield the same contribution to the Laplacian as long as they are properly normalized by the factor in front of $\theta_i$, as we did.

Synchronization, $\theta_i = \theta_j$ for all $i \neq j$, is a solution of Eq. (1) and we are interested in the effect of $\alpha$ on its stability. The system allows analytical treatment following the multiorder Laplacian approach introduced in ref. [39]. We define the second-order Laplacian as

$$L_{ij}^{(2)} = k_i^{(2)} \delta_{ij} - \frac{1}{2} A_{ij}^{(2)}, \quad (3)$$

which is a natural generalization of the graph Laplacian $L_{ij}^{(1)} \equiv L_{ij} = k_i \delta_{ij} - A_{ij}$. Here, we used the generalized degree $k_i^{(2)} = \frac{1}{2} \sum_{j,k=1}^{n} B_{ijk}$ and the second-order adjacency matrix $A_{ij}^{(2)} = \sum_{k=1}^{n} B_{ijk}$.

Using the standard linearization technique, the evolution of a generic small perturbation $\delta\boldsymbol{\theta} = (\delta\theta_1, \cdots, \delta\theta_n)$ to the synchronization state can now be written as

$$\delta\dot{\theta}_i = -\sum_{j=1}^{n} L_{ij}^{(\text{mul})} \delta\theta_j, \quad (4)$$

in which the multiorder Laplacian is defined as

$$L_{ij}^{(\text{mul})} = \frac{1 - \alpha}{\langle k^{(1)} \rangle} L_{ij}^{(1)} + \frac{\alpha}{\langle k^{(2)} \rangle} L_{ij}^{(2)}. \quad (5)$$

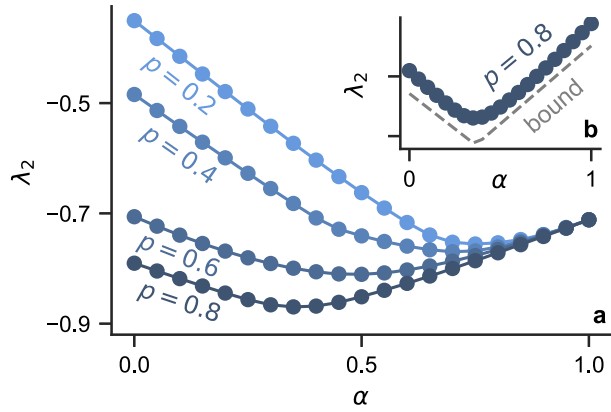

**Fig. 2 | Pairwise and non-pairwise interactions synergize to optimize synchronization. a** U-shaped curves are observed for $\lambda_2(\alpha)$ corresponding to random hypergraphs over a wide range of $p$ values. **b** Degree-based bound $|\lambda_2| \leq \frac{n}{n-1} k_{\min}$ predicts the non-monotonic dependence on $\alpha$. Each data point represents a 100-node random hypergraph and the three-body connection probability is set to $p_\triangle = 0.05$.

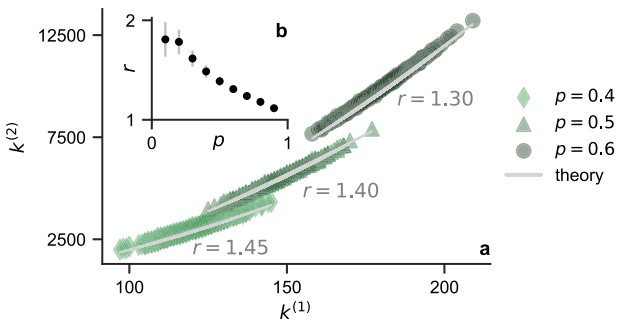

**Fig. 3 | Higher-order interactions increase degree heterogeneity in simplicial complexes. a** First-order degrees $k^{(1)}$ and second-order degrees $k^{(2)}$ follow Eq. (6) for simplicial complexes constructed from Erdös-Rényi graphs. The heterogeneity ratio $r$ is well approximated by Eq. (8). **b** Degree heterogeneity of the three-body couplings is larger than that of the pairwise couplings in simplicial complexes for all values of $p$ (i.e., $r$ is always greater than 1).

We then sort the eigenvalues of the multiorder Laplacian $\Lambda_1 \geq \Lambda_2 \geq \ldots \geq \Lambda_{n-1} \geq \Lambda_n = 0$. The Lyapunov exponents of Eq. (4) are simply the opposite of those eigenvalues. We set $\lambda_i = -\Lambda_{n+1-i}$ so that $0 = \lambda_1 \geq \lambda_2 \geq \ldots \geq \lambda_n$. The second Lyapunov exponent $\lambda_2 = -\Lambda_{n-1}$ determines synchronization stability: $\lambda_2 < 0$ indicates stable synchrony, and larger absolute values indicate a quicker recovery from perturbations.

We start by showing numerically the effect of $\alpha$ (the proportion of coupling strength assigned to second-order interactions) on $\lambda_2$. By considering the two classes of structures shown in Fig. 1: simplicial complexes and random hypergraphs, we find that these two canonical constructions exhibit opposite trends.

The construction of random hypergraphs is determined by wiring probabilities $p_d$: a $d$-hyperedge is created between any $d+1$ of the $n$ nodes with probability $p_d$[64]. Here, we focus on $d$ up to 2, so the random hypergraphs are constructed by specifying $p_1 = p$ and $p_2 = p_\triangle$. Simplicial complexes are special cases of hypergraphs and have the additional requirement that if a second-order interaction $(i, j, k)$ exists, then the three corresponding first-order interactions $(i, j)$, $(i, k)$, and $(j, k)$ must also exist. We construct simplicial complexes by first generating an Erdös-Rényi graph with wiring probability $p$, and then adding a three-body interaction to every three-node clique in the graph (also known as flag complexes).

Figure 1 shows that higher-order interactions impede synchronization in simplicial complexes, but improve it in random hypergraphs.

For simplicial complexes, the maximum transverse Lyapunov exponent $\lambda_2$ increases with $\alpha$ for all $p$ (data shown for $p = 0.5$ in Fig. 1). For random hypergraphs, the opposite monotonic trend holds for $p \simeq p_\triangle$. For $p$ significantly larger than $p_\triangle$, the curve becomes U-shaped, with the minimum at an optimal $0 < \alpha^* < 1$, as shown in Fig. 2.

Our findings also hold when we control the simplicial complex and random hypergraph to have the same number of connections (see Supplementary Fig. S1) and for simplicial complexes obtained by filling empty triangles in random hypergraphs[33], as shown in Supplementary Fig. S2. We note that instead of filling every pairwise triangle in the graph, we can also fill the triangles with a certain probability. As long as the probability is not too close to zero, the results in the paper remain the same (see Supplementary Fig. S3). One can also construct simplicial complexes from structures other than Erdös-Rényi graphs, such as small-world networks[65]. The results above are robust to the choice of different network structures. In Supplementary Figs. S4 and S5, we show similar results for simplicial complexes constructed from more structured networks, including small-world and scale-free networks.

### Linking higher-order representation, degree heterogeneity, and synchronization performance

To gain analytical insight on synchronization stability, we note that the extreme values of the spectrum of a Laplacian can be related to the extreme values of the degrees of the associated graph: $\lambda_n$ can be bounded by the maximum degree $k_{\max}$ from both directions, $\frac{n}{n-1} k_{\max} \leq |\lambda_n| \leq 2k_{\max}$[66]; and $\lambda_2$ can be bounded by the minimum degree $k_{\min}$ from both directions, $2k_{\min} - n + 2 \leq |\lambda_2| \leq \frac{n}{n-1} k_{\min}$[67]. For the multiorder Laplacian, the degree $k_i^{(\mathrm{mul})}$ is given by the weighted sum of degrees of different orders, in this case $k_i^{(\mathrm{mul})} = \frac{1-\alpha}{\langle k^{(1)} \rangle} k_i^{(1)} + \frac{\alpha}{\langle k^{(2)} \rangle} k_i^{(2)} = L_{ii}^{(\mathrm{mul})}$. In Fig. 2, we show that $\frac{n}{n-1} k_{\min}$ is a good approximation for $|\lambda_2|$ in random hypergraphs and is able to explain the U-shape observed for $\lambda_2(\alpha)$.

These degree-based bounds allow us to understand the opposite dependence on $\alpha$ for random hypergraphs and simplicial complexes. For simplicial complexes, the reason for the deterioration of synchronization stability is the following: Adding 2-simplices to triangles makes the network more heterogeneous (degree-rich nodes get richer; well-connected parts of the network become even more highly connected), thus making the Laplacian eigenvalues (and Lyapunov exponents) more spread out.

To quantify this rich-get-richer effect, we focus on simplicial complexes constructed from Erdös-Rényi graphs $G(n, p)$. In this case, we can derive the relationship between the first-order degrees $k^{(1)}$ and second-order degrees $k^{(2)}$ (below we suppress the subscript $i$ to ease the notation when possible). If node $i$ has first-order degree $k^{(1)}$, then there are at most $\binom{k^{(1)}}{2}$ 2-simplices that can potentially be attached to it. For example, when node $i$ is connected to nodes $j$ and $k$, then the 2-simplex $\Delta_{ijk}$ is present if and only if node $j$ is also connected to node $k$. Because the edges are independent in $G(n, p)$, when the network is not too sparse, we should expect about $p\binom{k^{(1)}}{2}$ 2-simplices attached to node $i$:

$$k^{(2)} \approx \mathrm{E}[k^{(2)}] = p\binom{k^{(1)}}{2} = pk^{(1)}(k^{(1)} - 1)/2. \tag{6}$$

This quadratic dependence of $k^{(2)}$ on $k^{(1)}$ provides a foundation for the rich-get-richer effect. To further quantify how the degree heterogeneity changes going from the first-order interaction to the second-

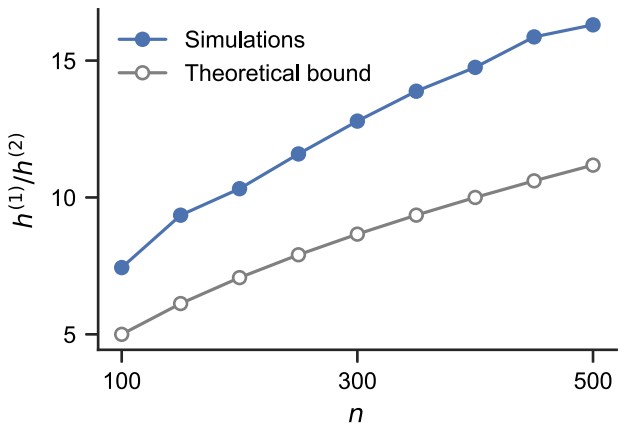

**Fig. 4 | Higher-order interactions decrease degree heterogeneity in random hypergraphs.** The degree heterogeneity (measured by $h^{(1)}$ and $h^{(2)}$) is stronger in the first-order Laplacian $\mathbf{L}^{(1)}$ than in the second-order Laplacian $\mathbf{L}^{(2)}$, and the difference increases with system size $n$. The theoretical lower bound of $\frac{h^{(1)}}{h^{(2)}}$ is given by $\frac{\sqrt{n}}{2}$ and is independent of $p$. The simulation results are obtained from random hypergraphs with different sizes $n$, using 500 samples for each $n$. The connection probabilities are set to $p = p_\triangle = 0.1$.

order interaction, we calculate the following heterogeneity ratio

$$r = \frac{k^{(2)}_{\max}/k^{(2)}_{\min}}{k^{(1)}_{\max}/k^{(1)}_{\min}}. \tag{7}$$

If $r > 1$, it means there is higher degree heterogeneity among 2-simplices than in the pairwise network, which translates into worse synchronization stability in the presence of higher-order interactions. Plugging Eq. (6) into Eq. (7), we obtain

$$r \approx k^{(1)}_{\max}/k^{(1)}_{\min} \geq 1. \tag{8}$$

This shows that the coupling structure of 2-simplices is always more heterogeneous than 1-simplices for simplicial complexes constructed from Erdös-Rényi graphs. Moreover, the more heterogeneous the pairwise network is, the greater the difference between first-order and second-order couplings in terms of heterogeneity. Specifically, because Erdös-Rényi graphs are more heterogeneous for smaller $p$, the heterogeneity ratio $r$ becomes larger for smaller $p$.

Figure 3a shows $k^{(1)}$ vs. $k^{(2)}$ for three simplicial complexes with $n = 300$ and various values of $p$. The relationship between $k^{(1)}$ and $k^{(2)}$ is well predicted by Eq. (6). The heterogeneity ratio $r$ is marked beside each data set and closely follows Eq. (8). Figure 3b shows $r$ as a function of $p$ for $n = 300$. The error bar represents the standard deviation estimated from 1000 samples. The data confirm our prediction that $r > 1$ for all considered simplicial complexes, and the difference between the first-order and second-order degree heterogeneities is most pronounced when the pairwise connections are sparse.

Next, we turn to the case of random hypergraphs and explain why higher-order interactions promote synchronization in this case (assuming that $p = p_\triangle$). For Erdös-Rényi graphs $G(n, p)$, the degree of each node is a random variable drawn from the binomial distribution $B(k; n, p) = \binom{n}{k} p^k q^{n-k}$, where $\binom{n}{k}$ is the binomial coefficient and $q = 1 - p$. There are some correlations among the degrees, because if an edge connects nodes $i$ and $j$, then it adds to the degree of both nodes. However, the induced correlations are weak and the degrees can almost be treated as independent random variables for sufficiently large $n$ (the degrees would be truly independent if the Erdös-Rényi graphs were directed). The distribution of the maximum degree for

large $n$ is given in ref. [68]:

$$P\left(k^{(1)}_{\max} < pn + (2pqn \log n)^{1/2} f(n, y)\right) \approx e^{-e^{-y}}, \tag{9}$$

where $f(n, y) = 1 - \frac{\log \log n}{4 \log n} - \frac{\log(\sqrt{2\pi})}{2 \log n} + \frac{y}{2 \log n}$.

For generalized degrees $k^{(2)}$, the degree correlation induced by three-body couplings is stronger than the case of pairwise interactions, but it is still a weak correlation for large $n$. To estimate the expected value of the maximum degree, one needs to solve the following problem from order statistics: Given a binomial distribution and $n$ independent random variables $k_i$ drawn from it, what is the expected value of the largest random variable $E[k_{\max}]$? Denoting the cumulative distribution of $B(N, p)$ as $F(N, p)$, where $N = (n - 1)(n - 2)/2$ is the number of possible 2-simplices attached to a node, the cumulative distribution of $k^{(2)}_{\max}$ is simply given by $F(N, p)^n$. However, because $F(N, p)$ does not have a closed-form expression, it is not easy to extract useful information from the result above.

To gain analytical insights, we turn to Eq. (9) with $n$ replaced by $N$, which serves as an upper bound for the distribution of $k^{(2)}_{\max}$. To see why, notice that Eq. (9) gives the distribution of $k^{(1)}_{\max}$ for $n$ (weakly-correlated) random variables $k^{(1)}_i$ drawn from $B(n, p)$. For $k^{(2)}_{\max}$, we are looking at $n$ random variables $k^{(2)}_i$ with slightly stronger correlations than $k^{(1)}_i$, now drawn from $B(N, p)$. Thus, Eq. (9) with $n$ replaced by $N$ gives the distribution of $k^{(2)}_{\max}$ if one had more samples ($N$ instead of $n$) and weaker correlations. Both factors lead to an overestimation of $E[k^{(2)}_{\max}]$, but their effects are expected to be small.

To summarize, we have

$$P\left(k^{(2)}_{\max} < pN + (2pqN \log N)^{1/2} f(N, y)\right) > e^{-e^{-y}}. \tag{10}$$

Solving $e^{-e^{-y_0}} = \frac{1}{2}$ gives $y_0 \approx 0.52$. Plugging $y_0$ into the left-hand side of Eq. (9) and (10) yields an estimate of the expected values of $k^{(1)}_{\max}$ and $k^{(2)}_{\max}$, respectively. (Note that here for simplicity, we use the median to approximate the expected value). Through symmetry, one can also easily obtain the expected values of $k^{(1)}_{\min}$ and $k^{(2)}_{\min}$. To measure the degree heterogeneity, we can compute the heterogeneity indexes

$$\begin{aligned} h^{(1)} &= \left(E[k^{(1)}_{\max}] - E[\overline{k^{(1)}}]\right) / E[\overline{k^{(1)}}], \\ h^{(2)} &= \left(E[k^{(2)}_{\max}] - E[\overline{k^{(2)}}]\right) / E[\overline{k^{(2)}}], \end{aligned} \tag{11}$$

which controls $\lambda_2$ through degree-based bounds. Here, the expected values of the mean degree are given by $E[\overline{k^{(1)}}] = pn$ and $E[\overline{k^{(2)}}] = pN$, respectively.

Now, how do the first-order and second-order degree heterogeneities compare against each other? Using Eq. (9) to (11), we see that

$$\frac{h^{(1)}}{h^{(2)}} > \frac{(2qn^{-1} \log n)^{1/2} f(n, y_0)}{(2qN^{-1} \log N)^{1/2} f(N, y_0)}. \tag{12}$$

For large $n$, we can assume $f(n, y_0) \approx f(N, y_0) \approx 1$ and simplify Eq. (12) into

$$\frac{h^{(1)}}{h^{(2)}} > \frac{(n^{-1} \log n)^{1/2}}{(N^{-1} \log N)^{1/2}} \approx \frac{\sqrt{n}}{2}. \tag{13}$$

First, note that $\frac{h^{(1)}}{h^{(2)}} > 1$ for almost all $n$, which translates into better synchronization stability in the presence of higher-order interactions. The scaling also tells us that, as $n$ is increased, the difference in degree

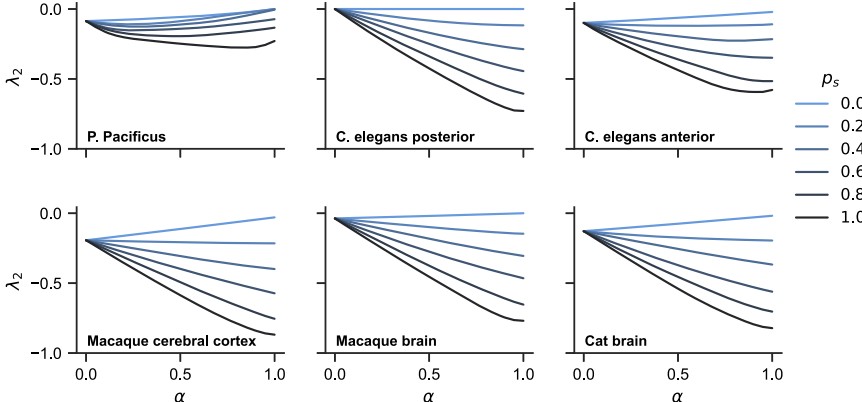

**Fig. 5 | Synchronization stability of hypergraphs constructed from brain networks.** For each of the six brain networks, we start by attaching 2-simplices to closed triangles and turning the network into a simplicial complex. We then construct hypergraphs at different distances from the simplicial complex by randomly shuffling the 2-simplices with probability $p_s$. For each value of $p_s$, we plot synchronization stability $\lambda_2$ as a function of the control parameter $\alpha$ (averaged over 50 independent realizations). It is clear that as the hypergraphs move further away from being a simplicial complex, synchronization stability is consistently improved. The role of higher-order interactions also generally transitions from impeding synchronization to promoting synchronization as $p_s$ is increased.

heterogeneities becomes more pronounced. The theoretical lower bound [Eq. (13)] is compared to simulation results in Fig. 4, which show good agreement. Intuitively, the (normalized) second-order Laplacian has a much narrower spectrum compared to the first-order Laplacian with the same $p$ because binomial distributions are more concentrated for larger $n$ (i.e., there is much less relative fluctuation around the mean degree for $k^{(2)}$ compared to $k^{(1)}$).

## Exploring the hypergraph space with synthetic networks and brain networks

So far we have focused mostly on simplicial complexes and random hypergraphs, which offered analytical insights into how higher-order representations influence collective dynamics. However, a vast portion of the hypergraph space is occupied by hypergraphs that are not simplicial complexes or random hypergraphs. What is the effect of higher-order interactions there? To explore the hypergraph space more thoroughly, we first construct simplicial complexes from both synthetic networks and real brain networks. We then study the synchronization stability of these structures as they move further and further away from being a simplicial complex. We find that as the distance to being a simplicial complex is increased, higher-order interactions quickly switch from impeding synchronization to promoting synchronization. This echoes the analytical results obtained above for simplicial complexes and random hypergraphs, and it supports our conclusion that higher-order interactions enhance synchronization in a broad class of (both structured and random) hypergraphs, except when they are close to being a simplicial complex.

Specifically, we consider the animal connectome data from Neurodata.io (https://neurodata.io/project/connectomes/), which consists of neuronal networks from different brain regions and different animal species. We chose brain networks because it has been shown that nonpairwise interactions and synchronization dynamics are both important in the brain[5,69]. For the sake of computational efficiency, we selected six networks spanning three different species (worm, monkey, and cat) that are neither too dense nor too sparse. Specifically, for each brain network, we first construct a simplicial complex by filling all 3-cliques (i.e., closed triangles) with 2-simplices. In reality, not all 3-cliques imply the existence of three-body interactions, and not all three-body interactions reside within 3-cliques. Thus, we continue by shuffling each 2-simplex to a random location in the hypergraph with probability $p_s$. This allows us to explore hypergraph structures beyond simplicial complexes and random hypergraphs, with $p_s$ also serving as a proxy for the distance between the hypergraph structure and the original simplicial complex.

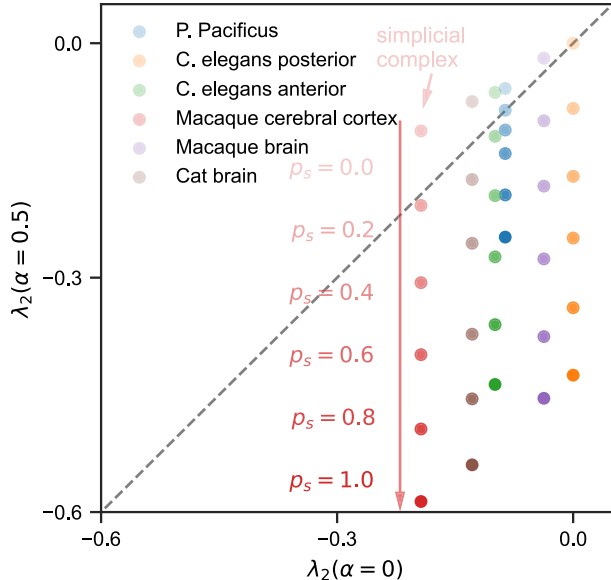

**Fig. 6 | Nonpairwise interactions typically enhance synchronization in hypergraphs, except when they are close to being a simplicial complex.** This plot uses the same data as in Fig. 5 and explicitly shows the transition in how higher-order interactions affect synchronization as the hypergraph structure loses resemblance to simplicial complexes. Points in the upper-left half of the square represent hypergraphs for which higher-order interactions impede synchronization, while points in the lower-right half of the square represent hypergraphs for which higher-order interactions promote synchronization. As the shuffling probability $p_s$ is increased ($p_s = 0$ corresponds to simplicial complexes), all six systems swiftly move across the diagonal line and into the synchronization-promoting region.

Figure 5 shows the synchronization stability $\lambda_2(\alpha)$ for hypergraphs constructed from the six brain networks at different values of shuffling probability $p_s$. Here, each curve represents an average $\lambda_2(\alpha)$ over 100 independent realizations of the hypergraph structure at a given $p_s$. We see that as $p_s$ is increased, for all systems, the curves change from going upward (or staying level for the disconnected *C. elegans* posterior network) with $\alpha$ to going downward with $\alpha$, signaling a transition from hyperedge-impeded synchronization to hyperedge-enhanced synchronization. Thus, the opposite trends we observed for simplicial complexes and random hypergraphs remain valid for a broad class of hypergraphs constructed from real-world networks.

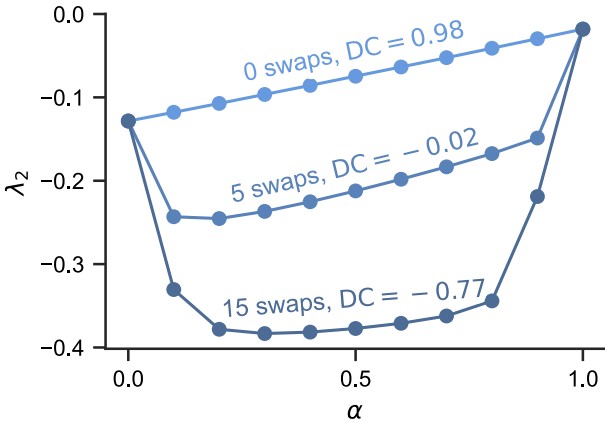

**Fig. 7 | Cross-order degree correlation affects synchronization stability in systems with mixed pairwise and nonpairwise interactions ($0 < \alpha < 1$), but not in the uniform cases ($\alpha = 0$ and $\alpha = 1$).** The hypergraphs are constructed from the cat brain dataset. For each curve, we indicate the number of pairs of nodes selected for the hyperedge membership swapping procedure (see text for details). The procedure changes the cross-order degree correlation (DC) without affecting the degree sequences (thus preserving the degree heterogeneity ratio as well). We can see that more negative cross-order degree correlation translates into better synchronization stability.

Figure 6 summarizes the same result from a different perspective by plotting $\lambda_2(\alpha = 0.5)$ against $\lambda_2(\alpha = 0)$ at different values of $p_s$ (other choices of the two $\alpha$ values give similar results). As $p_s$ is increased from 0 to 1 and the hypergraph structure moves further away from being a simplicial complex, we see all six systems transition from the upper-left half of the plot (higher-order interactions impeding synchronization) to the lower-right half of the plot (higher-order interactions promoting synchronization). Moreover, the transitions happen fairly rapidly, with all systems crossing the diagonal line at $p_s < 0.2$.

We also find similar results for hypergraphs constructed from synthetic networks including scale-free and small-world networks, which we show in Supplementary Figs. S6 and S7, and for real-world hypergraphs (Supplementary Fig. S8). One additional thing worth noting in Supplementary Figs. S6 and S7 are that as long as the network is not too sparse or dense, changing the network density mostly shifts all curves together in the vertical direction without affecting the transition from hyperedge-impeded synchronization to hyperedge-enhanced synchronization.

#### The role of degree correlation
Cross-order degree correlation, defined as the correlation between the degree vectors at each order, $DC = \text{corr}(\{k_i^{(1)}\}, \{k_i^{(2)}\})$, has been shown to affect epidemic spreading and synchronization, where it can promote the onset of bistability and hysteresis[70,71]. By construction, degree correlation is large and positive in simplicial complexes due to the inclusion condition and close to zero in random hypergraphs. Here, we investigate the effect of cross-order degree correlation on synchronization to provide a more complete picture of why higher-order representations matter.

To isolate the effects of degree correlation from those of degree heterogeneity, we propose a method to fix the latter while changing the former. Starting from a simplicial complex, we first select two nodes: a node $i$ included in only a few triangles (low $k_i^{(2)}$) and a node $j$ included in many triangles (large $k_j^{(2)}$). Then we swap their respective values of $k^{(2)}$ by swapping the 2-simplices to which node $i$ and node $j$ belong. The hyperedge membership swap has the expected effect: it lowers the degree correlation without changing the degree heterogeneity. The extent to which the swap lowers the correlation depends on the respective degrees of the nodes at each order. In

particular, a simple way to maximize this effect is to iteratively swap the nodes that have the lowest and the largest $k^{(2)}$. Note that this swapping procedure is different from the shuffling procedure used for Figs. 5 and 6 (shuffling does not preserve the second-order degree sequence).

In Fig. 7, we show the result of the hyperedge membership swap on the cat brain network. The starting simplicial complex is the one used in Fig. 5, for which we do not swap any memberships. Then, we build two hypergraphs from it by selecting 5 and 15 pairs of nodes and swapping their 2-simplices as described above. As a result, $\lambda_2$ is lowered for intermediate values of $\alpha$. Importantly, though, the endpoints $\lambda_2(\alpha = 0)$ and $\lambda_2(\alpha = 1)$ remain unchanged, since only one order of interactions is present. These observations are confirmed on hypergraphs constructed from the other five brain networks (Supplementary Fig. S9) and from synthetic networks (Supplementary Fig. S10).

To summarize, lowering the cross-order degree correlation can help improve the synchronization stability when there is a mixture of pairwise and nonpairwise interactions. Intuitively, this makes sense because negative correlation allows the degree heterogeneity from two different orders of interactions to compensate each other and homogenize the hypergraph structure.

## Discussion
To conclude, using simple phase oscillators, we have shown that higher-order interactions promote synchronization in a broad class of hypergraphs but impede it in simplicial complexes. We have identified higher-order degree heterogeneity and degree correlation as the underlying mechanism driving these opposite trends. Although we only considered two-body and three-body couplings, this framework naturally extends to larger group interactions.

Do the lessons learned here for phase oscillators carry over to more general oscillator dynamics? The generalized Laplacians used here have been shown to work for arbitrary oscillator dynamics and coupling functions[41]. Moreover, the spread of eigenvalues of each Laplacian carries critical information regarding the synchronizability of the corresponding level of interactions. Thus, once different orders of coupling functions have been properly normalized, we expect the findings here to transfer to systems beyond coupled phase oscillators. That is, for generic oscillator dynamics, higher-order interactions should in general promote synchronization if the hyperedges are more uniformly distributed than their pairwise counterpart. In the future, it would be interesting to generalize our results to systems with nonreciprocal interactions[72–75].

Finally, while here we focused on the synchronization of coupled oscillators, our results are likely to have implications for other processes. These include processes as different as diffusion[30], contagion[70], and evolutionary processes[36], in which degree heterogeneity and degree correlation play a key role, and yet the differences between simplicial complexes and hypergraphs have been mostly treated as inconsequential. All in all, our results suggest that simplicial complexes and hypergraphs cannot always be used interchangeably and future research should consider the influence of the chosen representation when interpreting their results.

## Data availability
The brain connectome data used in Figs. 5 to 7 can be found at https://neurodata.io/project/connectomes/. All other data needed to evaluate our results are present in the paper. Additional data related to this paper may be requested from the authors.

## Code availability
The code to reproduce the main results is available at https://github.com/maximelucas/HOI_shape_sync_differently or on Zenodo https://doi.org/10.5281/zenodo.7662113[76], and makes use of the XGI library[77].

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

## Acknowledgements

We thank Alessio Lapolla, George Cantwell, Giovanni Petri, Nicholas Landry, and Steven Strogatz for insightful discussions. Y.Z. acknowledges support from the Schmidt Science Fellowship and Omidyar Fellowship. M.L. acknowledges partial support from Intesa Sanpaolo Innovation Center.

## Author contributions

Y.Z., M.L., and F.B. conceived the research. Y.Z. and M.L. performed the research. Y.Z., M.L., and F.B. wrote the manuscript.

## Competing interests

The authors declare that they have no competing interests.
