## [Peer Review File · Nature Communications]

Reviewers' comments:

Reviewer #1 (Remarks to the Author):

The authors study how the representation of higher-order interactions affects the synchronization of identical oscillators. The topic is timely and there are some interesting results in the paper. However, I have concerns about the main message and the methodology. My main problem is that the authors make very general claims based on very specific choices. The main claim is that hyperedges typically enhance synchronization in hypergraphs but have the opposite effect on simplicial complexes. However, this claim is based on a particular choice of hypergraph construction, namely one in which two- and three-body interactions are placed randomly. One could also construct hypergraphs by allocating three-body interactions to nodes which already have two-body interactions without making it a simplicial complex (in fact, without the three-body interactions including any of the two-body interactions for a given node). Synchronization on such a hypergraph, which also has degree heterogeneity, could behave like it does for simplicial complexes. In short, the behaviour that the authors observe could be very dependent on how the hypergraph is constructed. This very important issue is not addressed in the manuscript. I think the broad claims made in the paper are not appropriate and can not recommend the paper for publication.

In addition, I have a few additional issues:

- In Figure 1, it would be more transparent to set the values of p and p_{Δ} so that the numbers of two- and three-way interactions in both cases are the same.
- How is the line for the bound in the inset of Fig. 2 calculated? Is k_{\min} calculated for each realization, or is a theoretical formula being used?

Reviewer #2 (Remarks to the Author):

Summary

In this paper, the authors study the Kuramoto model on higher-order networks with pairwise and three-body interactions. More specifically, they construct random hypergraphs and simplicial complexes and compare the value of the maximum transverse Lyapunov exponent as they tune the importance of pairwise versus higher-order interactions. They show that for random hypergraphs, higher weights on the three-body interactions (mostly) favor synchronization, while it is always detrimental for simplicial complexes. They then show and argue that this effect is mainly due to a higher degree heterogeneity (for three-body interactions) in simplicial complexes compared to hypergraphs.

General comments

I enjoyed reading the article: it is well written and technically sound. Their claim---that the choice of higher-order structure representation matters for dynamical processes---is also very interesting, especially considering that dynamics on higher-order networks is a hot topic right now. It could have a significant impact on the field.

However, I don't think the results presented constitute compelling evidence for their claim. They explain that higher-order interactions in simplicial complexes are detrimental to synchronization because of a higher (generalized) degree heterogeneity. However, this feature is caused by the construction mechanism: placing random edges and promoting triangles to simplex, compared to placing both edges and hyperedges at random. They even go to great lengths to show that their two random ensembles produce different levels of generalized degree heterogeneity.

But what if the networks were reconstructed in another way? What about "real" simplicial complexes/hypergraphs where the higher-order interactions are inferred from the data and not just

from pairwise relations? Since degree heterogeneity is a feature of the random ensemble and not of the representation itself, I don't think their results substantiate their claim.

Also, while the authors focus on the impact of generalized degree heterogeneity, they do not address the potential role of correlation between the pairwise and three-body degrees. Indeed, a node participating in a 2-simplex has a pairwise degree of at least 2, which is not the case in hypergraphs. These correlations are intrinsic to the simplicial complex representation and could have an important impact on the dynamics as well. A potentially interesting avenue would be to compare dynamics on randomized hypergraphs/simplicial complexes having the same pairwise and three-body degree sequences.

If the authors can provide compelling empirical evidence for higher (generalized) degree heterogeneity in simplicial complexes, and not just because of the reconstruction approach, or substantiate their claim---for instance using randomized higher-order networks as above---I could reconsider the manuscript for publication at Nature Communication.

Minor issues

- The authors claim that their results are robust to other choices of network structure (WS, scale-free, etc.) beyond just the ER graphs. I suggest providing supplemental figures showing this.
- Equation (6): $k^{(2)}$ is an expected degree, I would emphasize this with the notation. Also, equation (7) becomes ill-defined for sparse networks, because k_{\min} can be zero. Is this why the networks used in the figures are always dense? Also, I don't think this ratio is a good measure of heterogeneity/dispersion. Since $k^{(1)}$ is distributed according to a binomial, and $k^{(2)}$ is a binomial given $k^{(1)}$, I wonder if it is possible to estimate analytically the "shape" of the distribution for $k^{(2)}$ and assess its heterogeneity.
- I believe the quadratic dependence on $k^{(1)}$ of the average on $k^{(2)}$ is more of a sign of high correlation than of heterogeneity (although both are probably true in this case). Going back to one of my main points above, I suggest investigating more in-depth the role of correlations.

Below, we provide a point-by-point response to the Reviewers' comments.

Response to Reviewer 1

The authors study how the representation of higher-order interactions affects the synchronization of identical oscillators. The topic is timely and there are some interesting results in the paper.

We thank the Reviewer for finding the study timely and with interesting results.

However, I have concerns about the main message and the methodology. My main problem is that the authors make very general claims based on very specific choices. The main claim is that hyperedges typically enhance synchronization in hypergraphs but have the opposite effect on simplicial complexes.

However, this claim is based on a particular choice of hypergraph construction, namely one in which two- and three-body interactions are placed randomly. One could also construct hypergraphs by allocating three-body interactions to nodes which already have two-body interactions without making it a simplicial complex (in fact, without the three-body interactions including any of the two-body interactions for a given node). Synchronization on such a hypergraph, which also has degree heterogeneity, could behave like it does for simplicial complexes. In short, the behaviour that the authors observe could be very dependent on how the hypergraph is constructed. This very important issue is not addressed in the manuscript. I think the broad claims made in the paper are not appropriate and can not recommend the paper for publication.

We thank the Reviewer for raising this important issue. In the original manuscript, in order to obtain analytical insights, we mostly focused on simplicial complexes and random hypergraphs. We fully agree with the reviewer that there are other ways to construct hypergraphs that should also be explored.

Inspired by the Reviewer's comment, we have explored the hypergraph space more systematically by studying a broad class of structured and semi-structured hypergraphs. In particular, we extended our analysis beyond random hypergraphs and simplicial complexes to include hypergraphs constructed from synthetic networks as well as hypergraphs inferred from real brain connectome data. We also systematically investigated the transitions from simplicial complexes to these hypergraph structures. We found that group interactions in simplicial complexes always impede synchronization, but they quickly change to promote synchronization as the hypergraph structure moves away from being a simplicial complex. These results support our key message that "hyperedges typically enhance synchronization in hypergraphs but have the opposite effect on simplicial complexes" and are reported in a new subsection in the main text along with Figs. 5, 6, S6-S8.

We also explored additional ways to construct random simplicial complexes and random hypergraphs, now shown in supplementary figures S1-S3. In all cases, the trends confirm the message of Figure 1.

We agree with the Reviewer that there are ways to construct hypergraphs with high degree heterogeneity and high cross-order degree correlation that are not simplicial complexes. To achieve the construction process described by the Reviewer, we have applied a configuration model on the 2-hyperedges: this shuffles them randomly while preserving (as well as possible) the second-order degree sequence. Results are shown in Fig. 1 of this document. That being said, simplicial complexes is the most natural (and most widely adopted) higher-order representation that can produce such highly heterogeneous hypergraph structures. Besides, simplicial complexes not

Figure 1: Configuration model on 2-hyperedges.

only represent a particular data point in the wider hypergraph class. They are also the natural way to encode topological information about relation data, different from hypergraphs which encode combinatorial information. This further supports our focus on simplicial complex vs hypergraphs.

In addition, I have a few additional issues: - In Figure 1, it would be more transparent to set the values of p and p_{Δ} so that the numbers of two- and three-way interactions in both cases are the same.

We have now added a supplementary figure S1 where we keep the number of 1- and 2-hyperedges the same between the hypergraphs and simplicial complexes by simply shuffling the 2-hyperedges, and it shows the same trends as in Figure 1.

- How is the line for the bound in the inset of Fig. 2 calculated? Is k_{min} calculated for each realization, or is a theoretical formula being used?

In Fig. 2, each curve corresponds to a single realization, for which we computed k_{min} . We then computed the bound as $(n/(n-1))k_{min}$, as now clarified in the caption.

Response to Reviewer 2

Summary— In this paper, the authors study the Kuramoto model on higher-order networks with pairwise and three-body interactions. More specifically, they construct random hypergraphs and simplicial complexes and compare the value of the maximum transverse Lyapunov exponent as they tune the importance of pairwise versus higher-order interactions. They show that for random hypergraphs, higher weights on the three-body interactions (mostly) favor synchronization, while it is always detrimental for simplicial complexes. They then show and argue that this effect is mainly due to a higher degree heterogeneity (for three-body interactions) in simplicial complexes compared to hypergraphs.

General comments— I enjoyed reading the article: it is well written and technically sound. Their claim—that the choice of higher-order structure representation matters for dynamical processes—is also very interesting, especially considering that dynamics on higher-order networks is a hot topic right now. It could have a significant impact on the field.

We thank the reviewer for appreciating our manuscript.

However, I don't think the results presented constitute compelling evidence for their claim. They explain that higher-order interactions in simplicial complexes are detrimental to synchronization because of a higher (generalized) degree heterogeneity. However, this feature is caused by the construction mechanism: placing random edges and promoting triangles to simplex, compared to placing both edges and hyperedges at random. They even go to great lengths to show that their two random ensembles produce different levels of generalized degree heterogeneity.

But what if the networks were reconstructed in another way? What about "real" simplicial complexes/hypergraphs where the higher-order interactions are inferred from the data and not just from pairwise relations? Since degree heterogeneity is a feature of the random ensemble and not of the representation itself, I don't think their results substantiate their claim.

We thank the reviewer for these important comments. We originally focused on simplicial complexes and random hypergraphs in order to obtain analytical insights on the effect of higher-order representations. We fully agree with the reviewer that there are other ways to construct hypergraphs that could also be explored to strengthen our claim.

To address this question, we have explored the hypergraph space more systematically by studying a broad class of structured and semi-structured hypergraphs.¹ In particular, we extended our analysis beyond random hypergraphs and simplicial complexes to include hypergraphs constructed from synthetic networks as well as hypergraphs inferred from real hypergraph and brain connectome data. We also systematically investigated the transitions from simplicial complexes to these hypergraph structures. We found that group interactions in simplicial complexes always impede synchronization, but they quickly change to promote synchronization as the hypergraph structure moves away from being a simplicial complex. These results support our key message that "hyperedges typically enhance synchronization in hypergraphs but have the opposite effect on simplicial complexes" and are reported in a new subsection in the main text along with Figs. 5, 6, S6-S8.

Also, while the authors focus on the impact of generalized degree heterogeneity, they do not address the potential role of correlation between the pairwise and three-body degrees. Indeed, a node participating in a 2-simplex has a pairwise degree of at least 2, which is not the case in hypergraphs. These correlations are intrinsic to the simplicial complex representation and could have an important impact on the dynamics as well. A potentially interesting avenue would be to compare dynamics on randomized hypergraphs/simplicial complexes having the same pairwise and three-body degree sequences.

We thank the Reviewer for raising this very interesting point. When describing the "rich-gets-richer" effect in simplicial complexes, we implicitly had correlations in mind. To disambiguate their effect, we have now implemented a method that can change the cross-order degree correlation while keeping the degree heterogeneity fixed. The basic idea is the following: Consider the degree sequences at each order, $\{k_i^{(1)}\}$ and $\{k_i^{(2)}\}$. The cross-order degree correlation is simply the correlation of those two vectors and the degree heterogeneity at each order can be thought of as the variability inside the corresponding vector. We can then select two nodes—one with high $k^{(2)}$ and one with low $k^{(2)}$ —and swap their triangles memberships. This operation shuffles the vector $\{k_i^{(2)}\}$ while keeping the vector $\{k_i^{(1)}\}$ fixed. As a result, we can decrease or increase cross-order degree correlation without affecting the degree heterogeneity.

¹We decided to not systematically explore real-world hypergraph data, where group interactions are measured directly and not inferred from pairwise interactions, since they are currently still very scarce and are mostly concentrated in social systems (in which it would be a stretch to consider dynamics such as synchronization). We have however considered two social-contact hypergraph datasets, which also support our main conclusions.

By applying this method to simplicial complexes constructed from synthetic and brain networks, we found that $\lambda_2(\alpha)$ stays fixed at $\alpha = 0$ and $\alpha = 1$ as the cross-order degree correlation is changed from positive to negative, while the full curve becomes more and more U-shaped, with negative cross-order degree correlation significantly lowering $\lambda_2(\alpha)$ for intermediate α values (see the new Figs. 7, S9, and S10). This is expected since only pairwise interactions are present at $\alpha = 0$ and only three-body interactions play a role at $\alpha = 1$. For intermediate values of α , a negative cross-order degree correlation allows edges and hyperedges to compensate each other, homogenizing the overall degree distribution $\{k_i^{(\text{mul})}\}$ that controls the synchronization stability.

If the authors can provide compelling empirical evidence for higher (generalized) degree heterogeneity in simplicial complexes, and not just because of the reconstruction approach, or substantiate their claim—for instance using randomized higher-order networks as above—I could reconsider the manuscript for publication at Nature Communication.

As described in details above, in this revised version we report a more systematic investigation of the effect of higher-order degree heterogeneity across a variety of synthetic and real-world hypergraphs, which further strengthens our main conclusions.

Minor issues—

- The authors claim that their results are robust to other choices of network structure (WS, scale-free, etc.) beyond just the ER graphs. I suggest providing supplemental figures showing this.

We thank the Reviewer for this suggestion. We have now added results for simplicial complexes and hypergraphs constructed from small-world networks and scale-free networks in Figs. S4-S7.

- Equation (6): $k^{(2)}$ is an expected degree, I would emphasize this with the notation. Also, equation (7) becomes ill-defined for sparse networks, because k_{\min} can be zero. Is this why the networks used in the figures are always dense? Also, I don't think this ratio is a good measure of heterogeneity/dispersion. Since $k^{(1)}$ is distributed according to a binomial, and $k^{(2)}$ is a binomial given $k^{(1)}$, I wonder if it is possible to estimate analytically the “shape” of the distribution for $k^{(2)}$ and assess its heterogeneity.

We have now updated Eq. (6) to emphasize that the quadratic relation holds exactly only for the expected second-order degree. The Reviewer is right that Eq. (7) becomes ill-defined when $k_{\min} = 0$. However, we would like to point out that as long as the network is not extremely sparse (to the point that it becomes disconnected), the quadratic relation and Eq. (8) would hold regardless of the density of the network. For example, the inset of Fig. 3 shows results for p as low as 0.1. We agree that the heterogeneity ratio $r = \frac{k_{\max}^{(2)}/k_{\min}^{(2)}}{k_{\max}^{(1)}/k_{\min}^{(1)}}$ is not the only way to measure heterogeneity. However, it is simple and effective in this context, with the additional advantage that it can be directly linked to the degree-based bound we used to estimate λ_2 throughout the manuscript.

- I believe the quadratic dependence on $k^{(1)}$ of the average on $k^{(2)}$ is more of a sign of high correlation than of heterogeneity (although both are probably true in this case). Going back to one of my main points above, I suggest investigating more in-depth the role of correlations.

The Reviewer is correct in pointing out that the quadratic dependence of the expected value of $k^{(2)}$ on $k^{(1)}$ is a consequence of both high degree heterogeneity and high degree correlation—a

perfect cross-order degree correlation without the enhanced (second-order) degree heterogeneity would give us linear dependence, while an enhanced (second-order) degree heterogeneity without high cross-order degree correlation would destroy the functional relationship between $k^{(1)}$ and $k^{(2)}$. As mentioned above, we have now added systematic results on the role of degree correlation in a new subsection in the main text along with Figs. 7, S9, and S10.

REVIEWERS' COMMENTS

Reviewer #1 (Remarks to the Author):

The authors have addressed my concerns. Additional evidence to support their claims has been provided and the statements they made in the original version have been softened and made more specific. I think the paper can be published now.

Reviewer #2 (Remarks to the Author):

The new figures 5,6 and 7 give more depth to the analysis performed by the authors and provide more compelling evidence supporting their claim that higher-order interactions promote synchronization unless it is close to being a simplicial complex.

I like the progression toward a more targeted assessment of individual properties, culminating with figure 7 where only cross-order degree correlation is changed. My only comment would be related to figures 5 and 6: I would emphasize somewhere that in this case, the degree sequence of triangles is not preserved by the shuffling process, while it is preserved in figure 7.

Otherwise, the authors have addressed all my concerns. I now recommend the manuscript for publication.

Response to Reviewer 1

The authors have addressed my concerns. Additional evidence to support their claims has been provided and the statements they made in the original version have been softened and made more specific. I think the paper can be published now.

We want to thank the Reviewer again for his/her constructive comments that significantly improved our paper.

Response to Reviewer 2

The new figures 5,6 and 7 give more depth to the analysis performed by the authors and provide more compelling evidence supporting their claim that higher-order interactions promote synchronization unless it is close to being a simplicial complex.

I like the progression toward a more targeted assessment of individual properties, culminating with figure 7 where only cross-order degree correlation is changed. My only comment would be related to figures 5 and 6: I would emphasize somewhere that in this case, the degree sequence of triangles is not preserved by the shuffling process, while it is preserved in figure 7.

Otherwise, the authors have addressed all my concerns. I now recommend the manuscript for publication.

We want to thank the Reviewer again for his/her constructive comments that significantly improved our paper. We have now added one sentence emphasizing the difference between the shuffling and swapping procedures in Page 9.